# A second-generation molecular clamp stabilised bivalent candidate vaccine for protection against diseases caused by respiratory syncytial virus and human metapneumovirus

Andrew Young[1,2], Sharada Kolekar[1], Carlos Alvarez Mendoza[1], Noushin Jaberolansar[1,2], Naphak Modhiran[2], Tim Webb[1], Robert McCuaig[1], Varsha Kommajosyula[1], Nicolas Tardiota[1], Quimbe Dy[1], Alberto A. Amarilla[2], Rhiannon L. Dalrymple[1,2], Marianne Gillard[2], Julie L. Dutton[1,2,3], Juana Magdalena[3], Frank Vandendriessche[3], Jean Smal[3], Paul R. Young[1,2,4], Daniel Watterson[1,2,4], Emmanuel J. Hanon[3]*, Keith J. Chappell [1,2,3,4]*

1 The Australian Institute for Bioengineering and Nanotechnology, The University of Queensland, St Lucia, Queensland, Australia, 2 School of Chemistry and Molecular Biosciences, The University of Queensland, St Lucia, Queensland, Australia, 3 Vicebio Pty Ltd, Louvain-la-Neuve, Belgium, 4 Australian Infectious Diseases Research Centre, The University of Queensland, St Lucia, Queensland, Australia

* emmanuel@vicebio.com (EJH); k.chappell@uq.edu.au (KJC)

## Abstract

Respiratory syncytial virus (RSV) and human metapneumovirus (hMPV) are two medically important causes of respiratory tract infections and diseases. After more than five decades of research and development, vaccines have recently been approved for the prevention of lower respiratory tract disease caused by RSV. However, vaccines for hMPV remain in early-stage development. Here we describe the design and characterisation as well as pre-clinical development of a bivalent vaccine, VXB-241, comprised of the recombinantly expressed viral fusion proteins from both RSV and hMPV, stabilised in their pre-fusion conformation by combining the use of two technologies, the second-generation molecular clamp (MC2S) and key pre-fusion stabilizing mutations. Each of the two antigens were produced at high yield in a mammalian expression system and purified by an affinity capture resin specific to MC2S. Each antigen was demonstrated to adopt the pre-fusion conformation, which was stable for at least twelve months in liquid formulation at 2–8°C. Head-to-head evaluation in mouse immunogenicity studies showed that the VXB-241 candidate vaccine induced a neutralising immune response that was superior or equivalent to the pre-fusion stabilised comparator antigens for either RSV or hMPV, including the RSVPreF3 antigen of the licensed RSV vaccine, Arexvy (GSK). The results presented here have supported progression of VXB-241 into a Phase 1 clinical trial which commenced enrolment in August 2024 (ClinicalTrials.gov ID NCT06556147).

**Data availability statement:** All data are available within the manuscript and supporting information.

**Funding:** This work was supported by contract research funding provided by Vicebio Ltd to The University of Queensland researchers, PY, DW and KC. Vicebio representatives JM, FV, JS and EH contributed to project design, project administration, and review and editing of the manuscript. The funders had no role in study design, data collection and analysis, decision to publish, or preparation of the manuscript.

**Competing interests:** I have read the journal's policy and the authors of this manuscript have the following competing interests: Keith Joseph Chappell, Daniel Watterson and Paul Robert Young own intellectual property related to molecular clamp and shares in Vicebio. Keith Joseph Chappell, Julie Louise Dutton, Juana Magdalena, Frank Vandendriessche and Jean Smal are paid consultants for, and have shares/stock options in, Vicebio. Emmanuel Jules Hanon has shares/stock options and is the chief executive officer of Vicebio All other authors declare no competing interests.

## Author summary

A second-generation molecular clamp was used to help lock viral glycoproteins in a stabilised native prefusion conformation to elicit highly protective immunity and to enable consistent manufacture of highly stable, ready-to-use, fully liquid vaccines. The first- and second-generation molecular clamp platforms were previously validated in two successful Phase 1 clinical trials of SARS-CoV-2 vaccines (NCT04495933 and NCT05775887). This work is the first presentation of the pre-clinical development of a multi-pathogen vaccine, VXB-241 targeting both RSV and hMPV. The two antigens (VXB-213 and VXB-221) were shown to be stable for at least 12 months at 2–8°C and are able to elicit a strong neutralising immune response against RSV and hMPV in mouse immunogenicity study. VXB-241 is currently under evaluation in a first-in-human Phase 1 clinical trial (NCT06556147).

## Introduction

RSV was discovered in 1956 and causes infections at all ages and is a major cause of severe illness, especially for infants and older adults. There are 2 major antigenic subtypes of human RSV (A and B). By 2 years of age, virtually all children will have been infected with RSV. RSV is the most common cause of respiratory tract infections leading to bronchiolitis and pneumonia in children younger than 1 year of age in the United States. Moreover, the epidemiology of RSV shows high incidence of RSV hospitalisation among older patients as well as high risk for severe disease in immunocompromised adults [1]. Long-term care facility residents are particularly vulnerable to outbreaks and serious illness [2]. RSV burden of severe disease may be comparable to influenza [1].

Since the discovery of hMPV in 2001, the virus has been identified worldwide. Through genetic analysis hMPV has been characterised into 2 groups A and B which are further divided into 4 sub-lineages [3]. hMPV can cause upper and lower respiratory disease in people of all ages, especially among young children, older adults and people with weakened immune systems. hMPV is associated with acute lower respiratory infections and can result in bronchiolitis and pneumonia. Overall, the epidemiology and clinical manifestations of hMPV infection are like those in RSV infection [4]. Although nearly all populations will experience primary hMPV infection by age 5, hMPV re-infection occurs throughout adult life. Rates of hospitalisation of children for hMPV infection are highest in the first year of life but occur throughout early childhood. Pre-existing conditions, particularly asthma, play a role in disease severity and hospitalisation [5].

It has been documented that both RSV and hMPV have a high incidence rate and burden of illness, especially amongst susceptible populations including newborns, young children under 5 years of age, older adults, elderly and immunocompromised host. In children under five years of age, RSV infections were estimated to cause per

year 33 million acute lower respiratory infections (ALRIs), and to be associated to 3.6 million hospitalisations and 100,000 deaths worldwide [6]. Within the paediatric population globally in 2018, hMPV infection was estimated to be the cause per year of 11 million ALRI cases, 502,000 hospitalisations and 11,300 deaths [7]. In older adults over the age of 60, RSV infection is estimated to account for 5.2 million ALRIs, 470,000 hospitalisations and 33,000 deaths in high-income countries alone [8]. Within the older adult population, the burden of disease caused by hMPV has been less intensely studied, however there is an increased availability and implementation of molecular assays in clinical practice to analyse the virus causing lower respiratory tract disease [9]. Recent studies, conducted in the US and several countries worldwide, have shown that RSV and hMPV are similarly related to acute respiratory diseases, with similar reported percentage of virus detection, hospitalisations and deaths in susceptible populations [10].

After more than 5 decades of research and development [11,12], three vaccines have recently been approved in Europe, the USA and elsewhere for the prevention of lower respiratory tract disease (LRTD) caused by RSV in older adults: two subunit vaccines, Arexvy (GSK) [13] and Abrysvo (Pfizer) [14], and a mRNA vaccine, MRESVIA (Moderna) [15]. Abrysvo has also received approval for vaccination during pregnancy for the protection of infants against RSV throughout the first six months following birth [16]. Each vaccine has demonstrated high levels of protective efficacy and favourable safety in their respective target populations [13–16]. The transformational discovery at the heart of these advances has been the recognition that during the fusion of the viral envelope to the host cell membrane, glycoprotein F transitions irreversibly from a pre-fusion (RSV preF) to a post-fusion (RSV postF) conformation. RSV preF is an unstable conformation that is able to elicit strong neutralising antibody (nAb) activity while the conformationally more stable RSV postF elicits poor nAb activity [17,18].

While there are no vaccines currently available for hMPV, a bivalent mRNA vaccine developed by Moderna targeting hMPV and parainfluenza virus type 3 (PIV3) has demonstrated potential to boost neutralisation responses in a Phase 1 clinical trial [19]. Moderna is currently conducting further clinical trials of a bivalent mRNA vaccine targeting RSV and hMPV. Icosavax, now part of AstraZeneca, is also developing a bivalent RSV-hMPV vaccine (IVX-A12), based on virus-like particles (VLPs), which has been demonstrated to boost neutralsing antibody titres against both RSV and hMPV by 3–5 fold [20]. As with RSV, the hMPV preF has been demonstrated to elicit a higher neutralising immune response than hMPV postF [21–24], however there are also some differences between the two viruses. For RSV most neutralising immune responses are specific to RSV preF [17,18], however for hMPV neutralising immune responses overlap between hMPV preF and postF [25–27], with some neutralising antibodies (nAbs) recognising epitopes on the internal trimer interface [25,28].

While the newly approved RSV vaccines have great potential to protect vulnerable individuals and reduce the burden on hospitals during peak RSV seasons, equitable distribution of ready-to-use vaccine formulations remains a significant barrier to access these important vaccines, especially within LMICs. Both approved subunit vaccines (Arexvy and Abrysvo) are lyophilised products that require reconstitution prior to administration [13], and MRESVIA, like other mRNA vaccines, is stored frozen and must be thawed prior to administration, cannot be refrozen and must be discarded if not used within 24 hours of thawing [14]. Such requirements complicate real world use. Costly manufacturing and distribution requirements also likely add to the price per dose for these vaccines, impacting global supply and equitable access. A further challenge exists now that three separate respiratory virus vaccines are recommended for elderly populations (seasonal influenza, SARS-CoV-2 and RSV) as vaccination compliance becomes more challenging. The prospect of adding further vaccines to cover and protect against high burden respiratory diseases caused by other viruses such as hMPV and PIV into the current schedule will increase compliance challenges. To address this medical need, multivalent vaccines will be of increased importance and will require developing stable antigens that can be combined in one vaccine.

In this work, we describe the development and pre-clinical evaluation of a bivalent candidate vaccine (VXB-241) to provide protection against RSV and hMPV ALRIs. VXB-241 is composed of two antigens: RSV preF (VXB-213) and hMPV preF (VXB-221). The two antigens are stabilised in their pre-fusion conformation by combining the use of two

technologies, the proprietary second-generation, immuno-silenced, molecular clamp (MC2S) platform, which enables trimerisation and purification, and key pre-fusion stabilising mutations. The two candidate vaccine antigens show high production yield, liquid stability and the ability to elicit a strong neutralising immune response against RSV and hMPV. A first-in-human, Phase 1 clinical trial is currently underway with the key objective being the assessment of safety and immunogenicity of VXB-241 in healthy adults, 18–40 and 60–83 years of age (clinicaltrials.gov ID NCT06556147).

## Results

### Discovery and characterisation of the second-generation molecular clamp platform

Following the finding in late 2020 that the immune response induced by a first-generation molecular clamp (MC)-stabilised SARS-CoV-2 vaccine, Sclamp, interfered with some HIV diagnostic tests [29], the original MC platform was re-engineered to avoid the use of sequences derived from HIV-1 gp41. A panel of candidate heptad repeat sequences was screened to identify those that form highly stable 6HB structures. Subsequent rounds of optimisation and testing led to the identification of molecular clamp 2 (MC2); details available WO2023187743.

As HIV-1 gp41 derived sequences are not included in MC2, cross-reactivity to HIV diagnostics was not anticipated. However, given the similarities in quaternary structure, to rule this out an ELISA was conducted using two different reference serum samples readily available for HIV diagnostic testing. Testing via enzyme-linked immunosorbent assay (ELISA) showed that both NIBSC HIV reference serum and BioRad Geenius HIV+ control serum revealed reactivity to MC but no detectable reactivity to MC2 (S1 Fig).

Both MC and MC2 were compared, and tested with three different viral antigens that had been validated on the first-generation platform (antigen from SARS-CoV-2 [29], Nipah virus [30], influenza A virus [31]), as well as an early iteration of the RSV preF antigen that was in development. For the four antigens; comparable antigen homogeneity and oligomeric characteristics were conferred by MC and MC2, as demonstrated by size exclusion chromatography (SEC) (S2 Fig). Immunogenicity in mice was also compared, with sera from mice vaccinated with MC2 and MC2S stabilised antigens shown to elicit equivalent or superior levels of neutralising immune responses compared to MC (S3 Fig).

The second-generation clamp was further developed to incorporate additional N-linked glycosylation sites to dampen immunogenicity, referred to as Molecular Clamp 2 Silenced (MC2S). RSV preF incorporating MC2S was compared to MC and MC2, with antigen homogeneity again shown to be comparable and superior neutralising immune responses elicited in mice compared to MC (S2 and S3 Figs). The effect of silencing was demonstrated by comparing the immune response in mice after vaccination with RSV F antigen containing either MC, MC2 or MC2S. The relative percentage of elicited IgG binding to MC2S was reduced by >2-fold from a geometric mean of 34.3% for MC2 to 15.8% for MC2S (S4 Fig).

To further determine whether there was any cross-reactivity to HIV gp41, sera from mouse vaccinated with either MC or MC2 stabilised SARS-CoV-2 spike protein was analysed by ELISA against HIV gp41. The MC2 stabilised spike elicited approximately 38-fold lower cross-reactivity compared to MC, essentially at, or near the LoD of the assay (S5 Fig).

### Research-scale antigen production and characterisation

The RSV and hMPV candidate vaccine antigens including MC2S were generated and analysed side by side with well characterised recombinant protein controls for comparison. Briefly, there were two categories of controls, controls that were classed as non-stabilised soluble controls (RSV Fsol and hMPV Fsol), and controls that were preF stabilised benchmark comparators with a foldon trimerisation domain (RSV DsCav1 [32], hMPV DsCavEs2 [21] and hMPV v3B-Δ12_D454C-V458C [23]). From the new candidate vaccine antigens containing MC2S, one antigen has been selected for RSV, RSV preF, called VXB-213 and another for hMPV, hMPV preF, called VXB-221. VXB-213 was engineered to include the optimal disulfide bridge linkages and furin cleavage site deletion identified by McLellan et al., [32] and Joyce et al., [33], whereas VXB-221 included an unmodified TMPRSS2 cleavage site, a proline substitution and a single disulfide bridge [21,34]. The nomenclature with brief description of each antigen is summarised in Table 1.

**Table 1. Description of antigens and nomenclature.**

**RSV antigens**

| Name | Description | Ectodomain modifications | Trimer stabilising domain |
|---|---|---|---|
| RSV Fsol | Post-fusion control | Unmodified | N/A |
| DsCav1 | Pre-fusio n stabilised comparator [32] | Disulfide 155–290<br>S190F, V207L | Foldon (T4 fibritin) |
| VXB-213 | Selected (Vicebio) RSV preF candidate vaccine antigen | Deletion of aa 108–144<br>Disulfide 155–290<br>Disulfide 149–458<br>S190F, V207L, L373R | MC2S |

hMPV antigens

| Name | Description | Ectodomain modifications | Trimer stabilising domain |
|---|---|---|---|
| hMPV Fsol | Post-fusion control | Unmodified | N/A |
| DsCavEs2 | Pre-fusion stabilised comparator [21] | Furin RQSR>RRRR<br>Disulfide 365–463<br>Disulfide 127–153<br>Disulfide 140–147<br>Disulfide 110–322<br>A185P, L219K, V231I, E453Q | Foldon (T4 fibritin) |
| V3B-Δ12_D454C-V458C | Pre-fusion stabilised comparator [23] | - Furin deletion of aa89–112 replaced with GSGGSG<br>- Disulfide 84–147<br>- Disulfide 140–147<br>- Disulfide 454–458<br>- A185P | Foldon (T4 fibritin) |
| VXB-221 | Selected (Vicebio) hMPV preF candidate vaccine antigen | Disulfide 140–147<br>A185P | MC2S |

At a research laboratory scale for antigen discovery purposes, the recombinant antigens were expressed as secretable proteins in mammalian suspension culture (Chinese hamster ovary cells, ExpiCHO-S, Thermofisher) and purified from clarified supernatants by immunoaffinity chromatography using either a ligand-resin with affinity for MC2S (AVI-8740) or monoclonal antibodies (mAbs) targeting the viral ectodomain. They were buffer exchanged into PBS and characterised by size exclusion chromatography (SEC) using a Superdex Increase 10 300 GL (Cytiva) SEC column (Fig 1). Both the RSV and hMPV antigen panels showed similar trends for their oligomerisation profiles, with the non-stabilised Fsol proteins producing the greatest proportion of aggregate (RSV Fsol, 94% aggregate, 0% trimer; hMPV Fsol, 25% aggregate, 19% trimer, 56% monomer), the MC2S-incorporating antigens showing high degrees of trimer homogeneity (>89% trimer for both VXB-213 and VXB-221) with the comparator antigens being intermediate to these two (RSV DSCav1, 69% trimer; hMPV DSCavEs2, 54% trimer). hMPV DSCavEs2 produced two overlapping peaks within the trimeric fractions, suggesting that the trimer may adopt distinct conformational or oligomeric states. This observation differs from the results reported by Hsieh et al., [22], where hMPV DSCavEs2 was expressed in Freestyle 293-F cells, in the presence of co-transfected Furin and an additional SEC purification step. Similarly, we cannot exclude the possibility that RSV DSCav1 produced here in expiCHO-S (Thermofisher), may be conformationally different to that produced by McLellan et al., in HEK293K cells [32].

To determine whether the proteins were correctly folded, epitope presentation was tested by ELISA using panels of mAbs. For the RSV antigens, ELISAs were performed in a sandwich format with antigens captured using a reformatted murine IgG2 isotype of Motavizumab [35] and probed with preF-specific human-type mAbs D25 [36], MPE8 [37] and AM14 [38]. For VXB-213 (Vicebio RSV preF) and the DsCav1 comparator, sub-nanomolar dissociation constants ($K_D$) for all three mAbs were observed (Fig 1G). AM14 binds a preF, quaternary epitope and its binding to vaccine antigens has

been correlated with protective immunity [38]. This mAb showed poor reactivity to the non-stabilised Fsol antigen, while binding VXB-213 with 8 picomolar $K_D$; a 6-fold higher avidity relative to DsCav1.

For the hMPV antigens, a broader panel of mAbs targeting diverse epitopes across the topology of preF was tested against the antigen series in an indirect ELISA format (Fig 1H). VXB-221 (Vicebio hMPV preF) was shown to be recognised with sub-nanomolar binding to five of the six mAbs tested. Only DS7 [39], which binds to the post-fusion conformation or a pre/post intermediate, showed low affinity for VXB-221. Neutralising antibodies with moderate or strong pre-fusion preference (MPV467, MPV-487 and MPE8) [37,40], showed similar sub-nanomolar affinities for VXB-221 and comparator antigens DSCavEs2 and v3B-Δ12_D454C-V458C, but poor reactivity for hMPV Fsol. Neutralising antibodies known to recognise the internal trimer interface (MPV-458 and M8C10) [25,40], were able to bind with sub-nanomolar

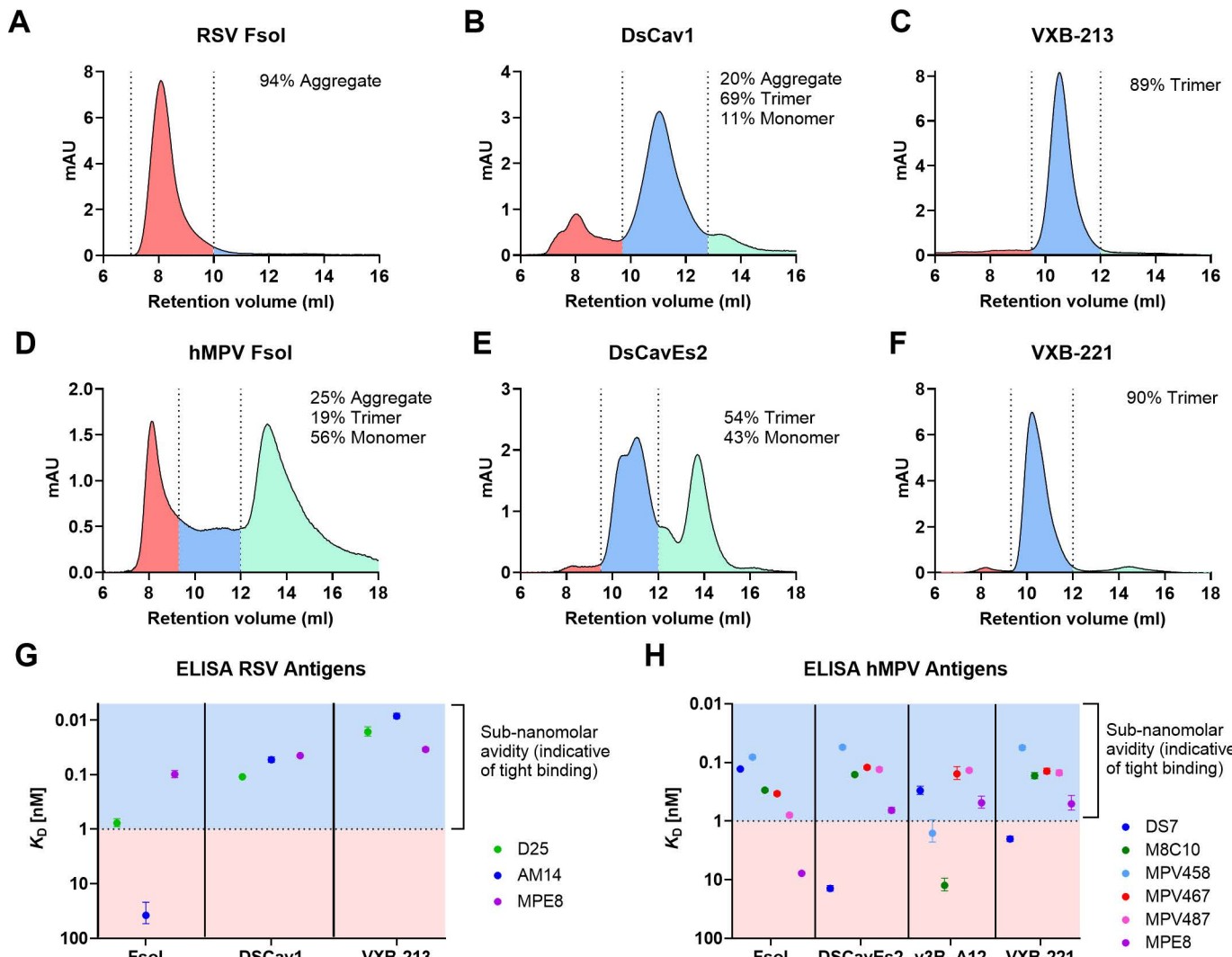

**Fig 1. *In vitro* characterisation of RSV preF and hMPV preF candidate vaccine antigens. (A-F)** SEC analysis of RSV and hMPV antigen panels using a Superdex 200 Increase 10/300 GL (Cytiva). The UV absorbance profiles in mAU are subdivided as 'aggregate' (red), 'trimer' (blue) and 'monomer' (green), with the percentage of total antigen calculated by area under curve (AUC). **(G-H)** mAb dissociation constant ($K_D$) for the RSV and hMPV antigens determined by ELISA and calculated by one site specific binding and plotted with SE. Analysis was performed using GraphPad Prism version 10.0.2 for Windows, GraphPad Software, San Diego, California USA, www.graphpad.com.

affinity to MPV-Fsol, DSCavEs2 and VXB-221 but showed poor affinity for v3B-Δ12_D454C-V458C, which contains an interprotomer disulphide bridge and has been optimised to adopt a closed, pre-fusion conformation [23].

Further analysis of the antigens by SDS-PAGE showed that the VXB-213 and VXB-221 were both highly pure and homogenous bands of predicted size for the non-cleaved products (S6 Fig). RSV Fsol and DsCav1 controls both showed a major band at the appropriate size for the furin cleaved product. hMPV Fsol showed a major band consistent with the non-cleaved product, DsCavEs2 also showed the presence of two major bands consistent with partial furin cleavage. All four controls exhibited some smearing that could be due to either impurities or proteolytic degradation.

The ability of each antigen to deplete neutralising antibodies from a pooled human plasma sample was compared either by directly pre-incubating plasma with antigens in solution or with immobilised antigens coupled to CNBR Activated Sepharose (Invitrogen). Consistent with previous reports [17,18], RSV Fsol performed poorly at removing neutralising antibodies from human plasma, whereas both DSCav1 and VXB-213, were highly efficient at depleting neutralising antibodies (S7A Fig). DSCavEs2 and VXB-221 were also both able to efficiently deplete hMPV neutralising antibodies, although hMPV Fsol was not tested (S7B Fig).

**Research-scale candidate vaccine immunogenicity and virus neutralisation in mouse**

BALB/c mice were immunised with 2 μg of the recombinant antigens by intramuscular injection (IM) in 2 doses, 3 weeks apart, with a squalene-based oil-in-water adjuvant, AddaVax (InvivoGen). To ensure RSV and hMPV Fsol antigens were fully in the post-fusion state they were first heated to 70°C for 2 hrs. At completion, analysis of mouse sera by enzyme-linked immunosorbent assay (ELISA) indicated that the immunisations had been effective in inducing IgG against each respective antigen. The bivalent candidate vaccine, VXB-241 (a combination of VXB-213 and VXB-221), elicited a response to both the RSV preF and hMPV preF antigens (Fig 2A). Each antigen also elicited a response to the trimer stabilising domain that was detectable via ELISA with a non-homologous antigen (Nipah F) stabilised with either foldon or MC2S. By comparing $EC_{50}$ values for total antigen-specific IgG and tag-specific IgG for each mouse we were able to estimate the proportion of tag-specific response. This response was considered undesirable as it will not contribute to protection. For VXB-213 and VXB-221 the geometric mean of the relative response to MC2S was found to be 9.3% and 6.3%, respectively, while with the bivalent VXB-241 the relative response to MC2S was found to be 4.7% (Fig 2B). In comparison, for the RSV F DSCav1 the geometric mean of the relative response to foldon was found to be 19.9%. For the hMPV F DSCavEs2 comparator, which also contains foldon, the geometric mean (GM) of the relative response to foldon was 3.1%. This response was somewhat variable with very low titres for 5 individual mice and relatively high titres for the remaining 3 mice.

These same sera were tested for neutralisation against RSV and hMPV by the plaque reduction neutralisation test ($PRNT_{50}$) (Fig 2C/D). This analysis demonstrates that VXB-213, at a dose of 2 μg delivered twice, 3 weeks apart, induces a neutralising immune response to RSV A2 (GM = 489; 95%CI = 222–1208). The response to VXB-213 is equivalent to comparator antigen RSV F DSCav1 (GM = 475; 95%CI = 255–946), and approximately 10-fold higher to heat-treated RSV Fsol (GM = 59; 95%CI = 32–125). The analysis of neutralisation against hMPV (Strain CAN97–83) demonstrates that VXB-221 induces a neutralising immune response to hMPV (GM = 1384; 95%CI = 1094–1762). The neutralising immune response elicited by VXB-221 was 2.5-fold higher than comparator antigen hMPV F DSCavEs2 (GM = 558; 95%CI = 336–971) and approximately 5.5-fold higher to heat-treated hMPV Fsol (GM = 252; 95%CI = 99–778). Statistical comparison of the neutralising immune response elicited by VXB-221 to comparator hMPV F DSCavEs2 showed a highly significant difference (Mann Whittney Test; p = 0.0003). The combined VXB-213 and VXB-221 (the selected RSV preF and hMPV preF) antigens mixed prior to injection in mouse, elicited a strong neutralising immune response against both RSV A2 (GM = 329; 95%CI = 171–687) and hMPV (Strain CAN97–83) (GM = 755; 95%CI = 543–1069).

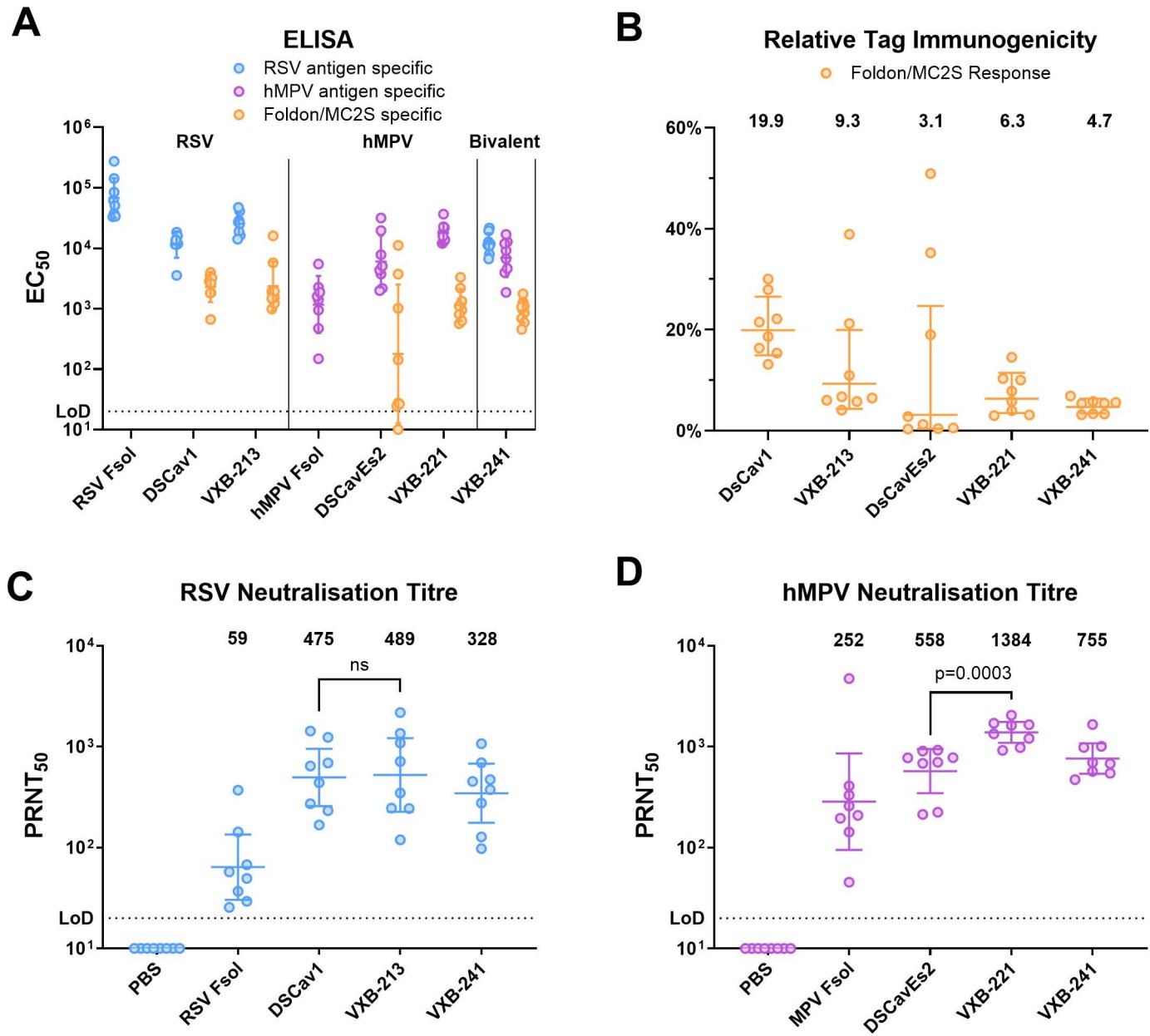

**Fig 2. Research-scale candidate vaccine immunogenicity in BALB/c mice. (A)** Serum IgG titre to the RSV F antigens (blue), hMPV F antigens (purple), and the IgG titre specific to either the foldon or MC2S trimer stabilising domain (orange). Titres were determined by ELISA and expressed as $EC_{50}$ values. **(B)** Relative proportion of foldon/MC2S reactivity calculated as a percentage of the total antigen specific titre. **(C/D)** Serum virus neutralisation titre determined in a plaque reduction neutralisation test ($PRNT_{50}$) against RSV A2 and hMPV (CAN97-83). Geometric means with geometric SD are plotted. Mann Whitney Test was performed to assess the statistical significance of differences between VXB-213 and comparator DSCav1 and between VXB-221 and comparator DSCavEs2. The limit of detection (LoD) is defined as half of the initial dilution factor used in the titration and is indicated with a dotted line. Analysis was performed using GraphPad Prism version 10.0.2 for Windows, GraphPad Software, San Diego, California USA, www.graph-pad.com.

## A 50L scale expression and characterisation of VXB-213 and VXB-221

A larger scale 50L manufacturing scale production run was performed with the resulting antigens analysed and data compared. For this 50L scale production, stably transfected pools were generated for each antigen. High expression clonal cell lines for VXB-213 and VXB-221 were isolated from CHO K1 stable cell pools and a top clone was selected for production of two specific clonal cell banks. For each antigen an initial production batch was completed at 50L scale to generate material for use in the currently described mouse immunogenicity studies.

Production of VXB-213 yielded 1.7 g/L and production of VXB-221 yielded 5.5 g/L respectively compared to laboratory scale transient production using the ExpiCHO-S (Thermofisher). This corresponds to production level increases of 70- and 300-fold for VXB-213 and VXB-221, respectively.

To assess antigen stability in a liquid formulation over time, a panel of assays was developed. This panel included monitoring for proteolytic degradation by Capillary Electrophoresis under reducing and denaturing conditions (CE-SDS), monitoring for the stability of the soluble trimer by Size Exclusion High Performance Liquid Chromatography (SE-HPLC) and use of a capture ELISA based potency test for each antigen. For VXB-213 (RSV preF), the ELISA included the mAbs Motavizumab [41] to capture and AM14 [38] to probe and for VXB-221 (hMPV preF), DS7 [39] to capture and MPE8 [37] to probe. Stability assessment conducted over twelve months with VXB-213 and VXB-221, showed that both antigens retain >99% single product by CE-SDS, > 90% soluble trimer by SE-HPLC and no decrease in reactivity by Potency ELISA (S8 Fig).

Material from the RSV preF (VXB-213) 50L batch was biochemically tested by SDS-PAGE alongside the current Arexvy (GSK) and Abrysvo (Pfizer) antigens. As these comparator antigens are distributed as a lyophilised powder, they were first reconstituted in PBS containing 0.05% PS80. Under reducing conditions, the VXB-213 and VXB-241 samples were highly pure with >95% product forming a single band at 60–70 kDa (S7 Fig). Both the Arexvy and Abrysvo antigens produced a major band between the 37 and 50 kDa markers with faint smearing of larger molecular weight products, and a secondary band slightly larger than 20 kDa. This appears to be consistent with near complete furin cleavage of the antigens.

In addition to the above, the antigens manufactured at the 50L bioreactor scale were similarly tested for the induction of immunogenicity and neutralisation responses using BALB/c mice. This study involved two immunisations of either 0.5 µg or 3 µg of each antigen (RSVpreF3 from GSK's RSV vaccine Arexvy, VXB-213, and VXB-221), or with either 1 µg or 6 µg of the combined VXB-213 and VXB-221 mixed prior mouse injection, called here VXB-241. Two doses were given, 3 weeks apart, all adjuvanted with AddaVax (Invitrogen). Three weeks after the second dose, serum was collected and assayed by ELISA to determine total IgG against each antigen, and the 'tag-specific' reactivity using an antigen containing either the MC2S domain or the foldon domain with a heterologous ectodomain (Fig 3A). For VXB-213, VXB-221 and bivalent VXB-241, the geometric mean of the relative response to MC2S for each group ranged between 3.7 and 11.2% (Fig 3B). In comparison, for the Arexvy RSVPreF3 antigen the geometric mean of the relative response to foldon was around twice as high at 15.5 and 15.4%.

Analysis of the mouse sera collected at week 6 following administration of IM vaccinations for neutralisation against RSV is shown in Fig 3C. At the lower dose level of 0.5 µg of RSV antigen, monovalent VXB-213 and bivalent VXB-241 both stimulated a strong neutralising immune response to RSV A2 (GM=5,272; 95%CI=2,884–10,046 and geomean=8,453; 95%CI=4,966–14,894, respectively), which was approximately 6- and 10-fold higher relative to the dose-matched Arexvy antigen (GM=817; 95%CI=420–1,710). At the higher dose level of 3 µg of the RSV antigen, monovalent VXB-213 and bivalent VXB-241 also stimulated a strong neutralising immune response to RSV A2 (GM=6,124; 95%CI=4,375–8,690 and geomean=7,534; 95%CI=5,082–8,690, respectively), which was approximately 2- and 2.5-fold higher relative to the dose matched Arexvy antigen (GM=2,958; 95%CI=1,361–6,982). Statistical comparison of the neutralising immune response elicited by VXB-213 to the response elicited by the Arexvy antigen showed a significant difference at the 0.5 µg dose level (Mann Whittney Test; p=0.003) but not at the higher dose level of 3 µg (Mann Whittney Test; p=0.13).

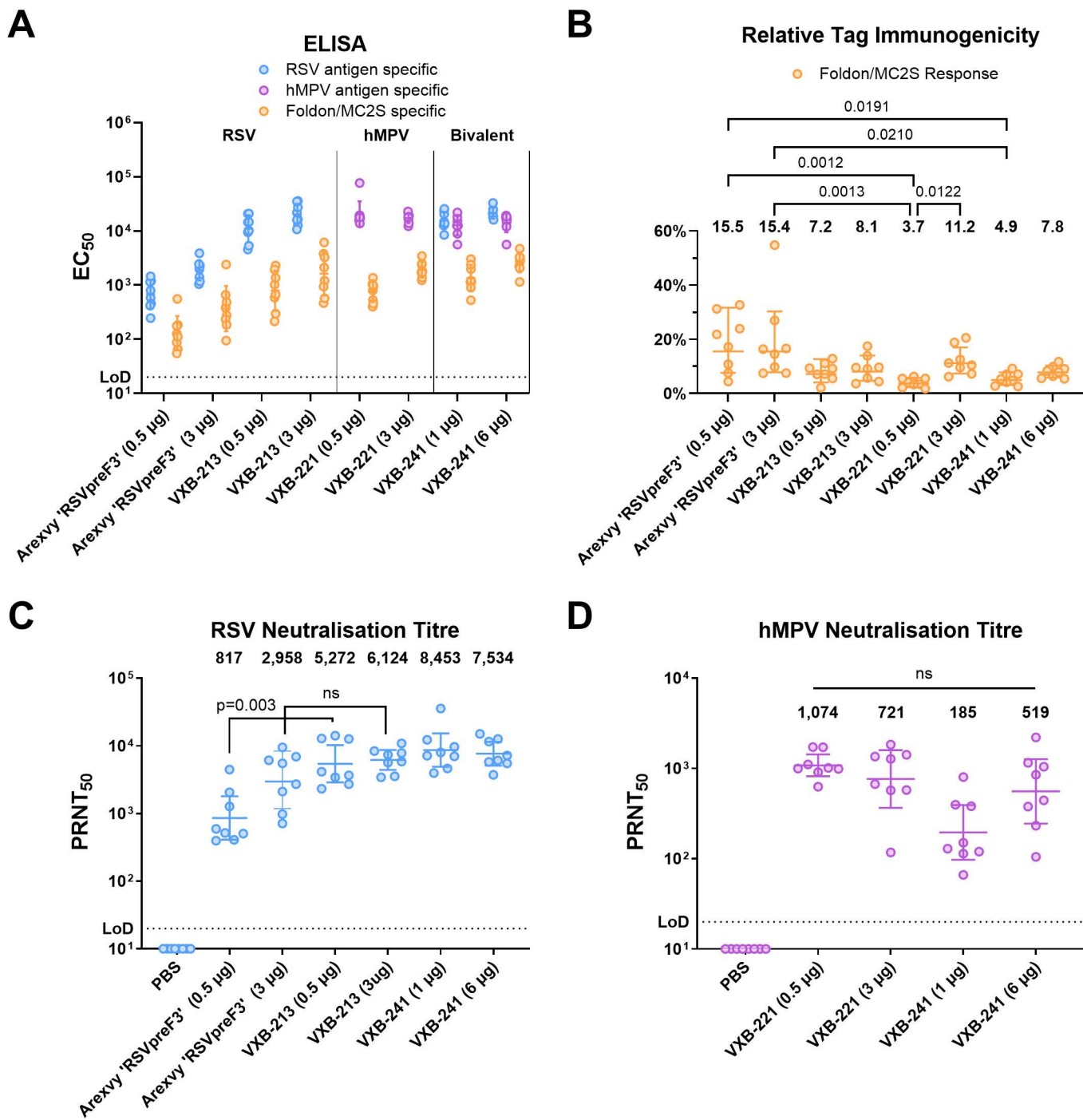

**Fig 3. Immunogenicity of 50 L scale antigen preparations in BALB/c mice. (A)** Serum IgG titre to VXB-213 (blue), VXB-221 (purple) or Nipah F containing either foldon or MC2S(orange). **(B)** The $EC_{50}$ values from the serum titrations are expressed as percentages of total IgG to show the proportion of total IgG directed towards the trimer stabilising domains. ANOVA Dunn's multiple comparisons test with only significant values >0.05 shown **(C)** Serum virus neutralisation titre ($PRNT_{50}$) determined against RSV A2. Mann Whitney Test was performed to assess statistically significant differences between VXB-213 and comparator RSVpreF3 antigen from Arexvy (GSK) at both the 0.5 and 3 µg dose levels. **(D)** Serum virus neutralisation titre ($PRNT_{50}$) determined against hMPV (strain CAN97-83). ANOVA Dunn's multiple comparisons test was performed to compare between four active groups and found to be non-significant. In A-D geometric means with geometric SD are plotted. The limit of detection (LoD) is defined as half of the initial serum dilution factor. Analysis was performed using GraphPad Prism version 10.0.2 for Windows, GraphPad Software, San Diego, California USA, www.graphpad.com.

PLOS Pathogens

The equivalent analysis of neutralisation against hMPV preF (Strain CAN97–83) is shown in Fig 3D. At the lower dose level of 0.5 µg of hMPV preF antigen, monovalent VXB-221 stimulated a strong neutralising immune response (GM = 1,074; 95%CI = 818 - 1,426), however the neutralising immune response was approximately 5-fold lower for the bivalent VXB-241 (GM = 185; 95%CI = 98–380), potentially indicating some interference due to immunodominance of the RSV preF antigen. At the higher dose level of 3 µg of hMPV antigen, both monovalent VXB-221 and bivalent VXB-241 stimulated a strong neutralising immune response (geomean = 721; 95%CI = 339 - 1,694 and geomean = 519; 95%CI = 235 - 1,288, respectively).

## Discussion

Historically, broadly applicable platform technologies for vaccines have saved countless lives. Such platform technologies have included bacterial polysaccharide vaccines (Meningococcus, Streptococcus, Hemophilus) [42], chemical inactivation of viruses (polio, influenza, rabies, Hepatitis A) and viral attenuation (Measles, Mumps, Rubella, Yellow Fever, Varicella) [43]. While successful, the application of these technologies to new targets often took upwards of 5->20 years. In recent years, a variety of platform technologies have been established that have significantly streamlined vaccine development and manufacturing. This is most clear in the mRNA vaccine revolution that has been brought forward in the context of the COVID-19 pandemic [44]. As there are still several challenges linked to the mRNA vaccines field (high reactogenicity, poor longevity, requirement for frozen storage) [44], having a broader variety of established vaccine platforms that are based on different approaches remains valuable. Here, we have demonstrated that vaccine antigens produced with the Molecular Clamp (MC) platform offer key advantages such as liquid stability and improved immune responses. The MC platform provides the versatility to target diverse viral pathogens.

The MC platform is essentially a high stability trimerization domain, similar to previously utilised alternatives GCN4 and Foldon [45,46]. The broadly applicable potential of a first-generation Molecular Clamp based on the HIV-1 gp41 6HB domain was demonstrated in preclinical studies for several viruses such as influenza, Ebola, Nipah, Lassa, and MERS coronavirus [30,31]. In 2020 a SARS-CoV-2 candidate vaccine progressed through Phase 1 clinical trial, with demonstrated safety and good immunogenicity profiles, but was not progressed due to interference with some HIV diagnostic tests, generating false positives [29]. A re-engineered, non-HIV antigen based MC2S sequence, provides the same candidate vaccine attributes, but overcomes the hurdle that prevented progression of MC.

The MC2S platform provides high production yield and purity with a universal purification system (AVI-8740 affinity resin), which facilitates rapid advancement through pre-clinical development. For two viral targets (RSV preF and hMPV preF) transient expression yields of 10 – 30 mg/L were achieved with purity above 95% by SDS-PAGE and approximately 90% homogenous trimer by SEC. This level of production comfortably accommodates pre-clinical testing, including *in vitro* assessment and animal immunisation studies. Furthermore, transition to 50L scale manufacture uses a comparable process regardless of antigen and was able to increase yields to >1 g/L. Production yields such as this with a consistent manufacturing process have the potential to dramatically reduce investment required to achieve commercial scale vaccine production.

Following the approval of a first generation of RSV vaccines, Arexvy (GSK) [13], Abrysvo (Pfizer) [14,16], MRESVIA (Moderna) [15], a large unmet medical need remains with the combined burden of other severe respiratory tract infections amongst highly susceptible populations including newborns, children under 5 years of age, older adult elderly and the immunocompromised. Moreover, there are areas in which the currently approved RSV vaccines can be improved. The two subunit vaccines (Arexvy and Abrysvo) are lyophilised products that require reconstitution prior to administration [13], and the LNP/mRNA vaccine (MRESVIA) is stored frozen and must be thawed prior to administration [14]. All three current RSV vaccines must be discarded if not used within a defined short period of time. In comparison, our current data show that the MC2S incorporating RSV antigen VXB-213, as well as VXB221 (hMPV preF), are both stable in liquid form at 2–8 °C for > 12 months. This finding supports compatibility with prefilled syringes at 2–8°C.

The approved RSV vaccines have been demonstrated to have good protective efficacy, which extends over seasons, however the dose level administered to achieve this protective efficacy is relatively high at 120 µg per dose. This is substantially higher than the 15 µg of hemagglutinin (HA) per strain included in standard dose influenza vaccines, and higher even than the 60 µg HA per strain included in high dose formulations [47]. We have shown in a dose and adjuvant matched naïve mouse immunogenicity study that VXB-213 elicited a significantly higher neutralising immune response than the Arexvy RSVpreF3 antigen at the lowest dose levels tested (p = 0.003), likely indicating better presentation of neutralising epitopes to naïve memory B-cells. This result is consistent with previous findings where a fully optimised RSV preF antigen which included the same pair of disulfide bridge stabilisations and optimal deletion within the furin cleavage site was shown to elicit strong neutralisating antibody responses relative to the original RSV preF DsCav1 antigen [33]. VXB-213 incorporates the same ectodomain modifications that are present in this optimised RSV preF antigen.

A further complexity exists now that three separate respiratory virus vaccines are recommended for elderly populations (for seasonal influenza virus, COVID-19 and RSV). hMPV is another high burden respiratory virus with a comparable incidence rate and severity to RSV in the older adult population [10]. The prospect of adding further vaccines into the annual schedule is a challenge and would likely impact coverage. We have focused on developing liquid stable antigens for both RSV and hMPV to be combined in a bivalent vaccine to reduce burden on healthcare providers and individuals.

While there are currently no approved vaccines for hMPV, there have been a wealth of pre-fusion stabilised antigens produced based on the learnings from RSV [21–24]. We chose two of these, DSCavEs2 [21] and v3B-Δ12_D454C-V458C [23], to include as comparators for in vitro analysis. Consistent with expectations, prefusion specific nAbs were able to bind with similar affinity to VXB-221 and comparator antigens, while nAbs recognising the internal trimer interface bound to VXB-221 and DSCavEs2 but not the closed pre-fusion stabilised antigen v3B-Δ12_D454C-V458C. DSCavEs2 was then included as a positive control within a dose and adjuvant matched naïve mouse immunogenicity study. Here, VXB-221 elicited a significantly higher neutralising immune response relative to HMPV preF DSCavEs2 (p = 0.0003), however it should be noted that differences in purification methodologies used here and those used by Hseih et al., [22] may have contributed to this difference. The 2.5-fold improvement in elicited neutralising immune response was unexpected given that DSCavEs2 has been highly optimised for stability of the pre-fusion conformation. Monoclonal antibody avidity to VXB-221 and DSCavEs2 also appeared to be relatively similar, except for neutralising mAb DS7 which recognises an epitope shared between the pre- and post-fusion conformations. A potential explanation for the improved neutralising immune response may relate to DSCavEs2 being optimised for efficient cleavage at the F1/F2 cleavage site [21], whereas VXB-221 is non-cleaved. It has previously been suggested that F1/F2 cleavage assists in the formation of a stable closed trimer, which was assumed, based on the earlier results for RSV preF would be the optimal antigen for stimulating a neutralizing immune response [23,24]. While nAbs recognising the internal trimer interface bound with similar affinity to both VXB-221 and DSCavEs2, accessibility to the trimer interface may be different in the context of vaccination. Alternatively, there may be additional neutralising epitopes that are specific to the non-cleaved hMPV preF. Of note, proteolytic cleavage of hMPV preF is inefficient relative to RSV preF [48,49], and timing of cleavage within the viral life cycle has not been well characterised. Contrary to structure-based antigen design focusing on the closed and/or fully cleaved hMPV preF which have included multiple structure stabilising mutations [21–24], VXB-221 is expected to adopt an open and non-cleaved preF conformation. This conformation allows for broad display of neutralising epitopes with minimal modification away from the native sequence. However, we note that monoclonal antibodies targeting the internal interface of the hMPV F protein exhibit varying levels of neutralizing potency. While some, such as M8C10 and MPV465, show modest neutralizing activity, others like MPV-458 demonstrate both high neutralizing potency and cross-reactivity across hMPV strains, highlighting the potential importance of this conserved epitope for vaccine design [28,50].

In mice that were vaccinated with the bivalent vaccine VXB-241, we observed some reduction in the neutralising immune response to both RSV and hMPV relative to the monovalent vaccines. This effect was particularly evident at the

lowest dosage tested, where the neutralising response to hMPV was reduced by 5-fold; this was not seen for RSV. This finding is consistent with the immunodominance of RSV F relative to hMPV F, however it is unknown whether this same phenomenon will be observed in human clinical trials.

As with the MC2S platform, approved RSV subunit vaccines, Arexvy (GSK) and Abrysvo (Pfizer) both include a preF stabilising trimerisation domain [13,14]. The trimer stabilising domain present in Arexvy and Abrysvo RSV preF antigen, is the T4 fibritin sequence-foldon based and the same foldon is also included in RSV preF DSCav1 and hMPV preF DSCavEs2 antigens [21,32]. It has recently been reported that foldon can stimulate an 'off-target response' in human vaccine recipients that may negatively impact vaccine potency after repeated administrations [51]. It was recently demonstrated that further stabilisation of the RSV preF ectodomain can facilitate the omission of a trimerization domain thereby eliminating any off-target immune response from this domain [51]. We also recognized the importance of reducing the off-target immune response and achieved this through the incorporation of multiple N-linked glycosylation sites into MC2S. This strategy provides the benefit of easy purification process without drawback of off target response. For each naive mouse immunogenicity study, we calculated the fraction of vaccine elicited IgG that was directed to the T4 fibritin based-foldon or the MC2S trimer stabilising domain, by directly coating antigen to ELISA plates. It should however be noted that direct coating can cause antigen denaturation and so such results may underestimate antigen specific or domain specific IgG responses. Nevertheless we found that typically 10–30% of IgG elicited to foldon containing antigens was specific to the foldon sequence, and therefore unable to contribute to virus neutralisation. This was highly variable and sometimes constituted 40–50% of the elicited IgG response. In comparison, the fraction of IgG directed to the MC2S domain was typically below 10% of the elicited IgG response. Even when mice were immunised with the bivalent VXB-241, in which both antigens contained an identical MC2S domain (with different ectodomains), there was very little overall IgG directed to MC2S. In an ongoing clinical trial by Vicebio, we are exploring the comparative immune responses against the trimerisation domains of both the MC2S containing candidate antigens, and Arexvy as an active comparator (NCT0655614).

In summary, the candidate vaccine antigens developed for RSV and hMPV using the second-generation, immuno-silenced, molecular clamp platform show high production yield, high liquid stability, and the ability to elicit a strong neutralising immune response against RSV and hMPV. A first-in-human, Phase 1 clinical trial is currently underway to assess safety and immunogenicity of VXB-241 in healthy adults, 18–40 and 60–83 years of age (clinicaltrials.gov ID NCT06556147). The molecular clamp platform technology contributes to the stabilisation of the antigens and offers the potential to advance the development of safe and effective vaccines for prevention of multiple medically important viral diseases. A subunit vaccine for SARS-CoV-2 generated with the non-immunosilenced, MC2 sequence, recently completed Phase 1 trial testing (clinicaltrials.gov ID NCT05775887) further demonstrating the broad applicability of the proprietary platform. The high yield and consistent production process of MC2S containing candidate vaccine antigens together with the focus on creating liquid stable, multivalent formulations, has the potential to maximise global distribution, improve vaccine compliance, and protect vulnerable populations from medically important respiratory viral diseases.

## Methods

### Ethics statement

All animal work related to this study was performed in accordance with policies for the proper use and care of animals set by The University of Queensland Animal Ethics Committee (AEC numbers: 2021/AE000123, 2021/AE000183, 2021/AE000571, 2023/AE000539). Experiments were designed to minimise animal numbers, and all animals were housed in suitable facilities with *ad libitum* access to food, water and environmental enrichment. All procedures were performed under anaesthesia by isoflurane inhalation by properly trained personnel. At the completion of the study, mice were euthanised by $CO_2$ asphyxiation.

## Antigen DNA vectors and small-scale protein expression

Vaccine antigen coding sequences codon optimised for *Cricetulus griseus* and synthetic DNA was synthesised in vitro as gblocks (IDT). Antigen coding sequences were cloned into mammalian expression vector pXC-17.4 (Lonza Group, Basel Switzerland) by In-Fusion Cloning (Clontech). The resulting plasmids were transformed into Stellar Competent Cells (Takara Bio) and screened for correct insertions by colony PCR. Plasmids from selected colonies were isolated and DNA sequencing was conducted to confirm sequence fidelity. Recombinant proteins were expressed transiently in ExpiCHO-S culture (ThermoFisher) using manufacturer's instructions.

RSV and hMPV antigens were purified from cell culture supernatant by affinity chromatography using either an affinity column coupled with monoclonal antibody 101F [52] or resin specific to the clamp domain (AVI8740, Avitide, Repligen). Additional antigens were purified by affinity chromatography with either MC (HIV-1 gp41) specific antibody, HIV1281 [53], SARS-CoV-2 specific antibody CR3022 [54], Nipah F virus specific antibody mab66 [55], or influenza H1 virus specific antibody 5J8 [56]. Immunoaffinity columns were prepared by chemical coupling of mAbs to NHS-Activated Sepharose Resing (Cytiva) using manufacturer's instructions. Antigens were purified using high pH elution from immunoaffinity resin as previously described [30,31]. Affinity purification of MC2S containing antigens was conducted by pre-equilibrating resin with 50mM Tris, 150mM NaCl, pH 7.4. The column was then washed with 50mM Tris, 600mM NaCl, pH 7.4 and eluted in 50mM Tris, 1M $MgCl_2$, pH7.4, before then being buffer exchanged into PBS pH 7.4. Homogeneity of purified proteins was analysed using SDS-PAGE and size exclusion chromatography.

## mAb generation

Panels of antibodies specific to RSV and MPV F proteins were generated by cloning the antibody variable domains into mab-xpress vectors by previously described methods [57]. The genetic constructs were expressed in CHO-S mammalian culture and purified using a MabSelect Sure affinity column (Cytiva). The variable domain sequences for 101F [52], D25 [36], MPE8 [37], AM14 [38], DS7 [39], MPV458, MPV483, MPV467, MPV487 [40], M8C10 [25], CR3022 [54], mab66 [55], and 5J8 [56] were obtained from public databases.

## In vitro protein screening, stability testing and lead identification

Purified proteins were characterised by standard ELISA using a panel of mAbs described above. Briefly, for RSV antigens ELISA 96-well Maxisorp plates (ThermoFisher) were coated with 2 µg/mL of mouse isotype Motavizumab (ref). Plates were blocked with blocking solution (PBS supplemented with 5% milk diluent (KPL, SeraCare)). Commercially acquired Arexvy (GSK) or Abrysvo (Pfizer) antigens were resuspended in PBS containing 0.05% PS80 and gently mixed to dissolve. Antigens were added to plates coated with capture antibody at 2 µg/mL in blocking buffer and incubated at 37°C for 1 hour. After washing with PBS-T, these plates were then probed with a serial dilution of the stated monoclonal antibody, starting at 33.33 nM and serially diluted 5-fold in blocking solution. After incubating for another hour at 37°C plates were washed as above and bound mAb was detected using goat anti human HRP (ThermoFisher) diluted 1:2000 in blocking buffer. TMB substrate was added and incubated at RT for 3 minutes 30 seconds before the reaction was stopped with 1M $H_2SO_4$. Absorbances were then read at 450 nm. Background absorbances of the full reaction without primary mAb were averaged and subtracted from the respective test absorbance readings.

Antigen oligomerisation profiles in solution were assessed by size exclusion chromatography (SEC) using a Superdex 200 Increase 10/300 GL resin filtration column (Cytiva). Antigens (40 ug) were applied using a 500 ul sample loop.

## 50L scale antigen production

50L batches of VXB-213 and VXB-221 were produced using each of the CHO K1 clonal cell lines to produce material for use in a human Phase 1 clinical trial. Briefly, cells were expanded in shake flasks and then a Rocking Motion Bioreactor

(Sartorius), before inoculation into 50L HyPerforma Single-Use Bioreactors (ThermoScientific). Production batches were maintained for 14 days with an optimised feed strategy and standard monitoring for growth rate, viable cell density, pH, osmolarity, glucose level, lactate level, ammonia level and carbon dioxide level

After the 14-day production run, cell culture supernatant was harvested by two-stage depth filtration and purified by AVI-8740 resin affinity capture, two ion exchange/mix mode column purification steps, viral inactivation with TNBP/PS80, viral nanofiltration and tangential flow filtration (TFF) into the final formulation buffer.

## Animal immunisations and sampling

Mouse immunisation studies were conducted using female BALB/c mice 6–8 weeks of age, housed at the Australian Institute for Bioengineering and Nanotechnology (AIBN) Animal Facility in the University of Queensland, Australia. Mice were immunised with 0.5 µg, 2 µg or 3 µg per recombinant protein antigen stated (maximum dose 6 µg for bivalent group) with squalene-based oil-in-water emulsion adjuvant, AddaVax (InvivoGen). Immediately prior to mouse vaccination, the Arexvy antigen vial was resuspended in PBS containing 0.05% PS80 and then mixed with AddaVax (InvivoGen).

Each group (n = 8) was bled via the caudal vein 1 day prior to receiving the first dose (day -1). They were then immunised IM on day 0 and again on day 21 (3 weeks). On day 42 (6 weeks), animals were euthanised and blood was taken by cardiac puncture. The blood was allowed to coagulate overnight at 4°C before centrifugation to isolate the serum. Serum was heat inactivated at 56°C for 30 minutes for use in immunological assays.

## Measuring antigen-specific IgG responses

Mouse IgG titres were determined by standard ELISA as described above with the following deviations. Antigens were coated directly on the ELISA plates at 2 µg/mL and probed with a 1/20 initial dilution of mouse serum that was serially diluted 5-fold in blocking solution. Following incubation and washing, the bound serum was detected using goat anti mouse HRP (ThermoFisher) at 1:2,000 dilution, before detection and reading as stated above. To determine the total IgG, the serum was tested against the vaccine antigens. To identify 'tag-specific' reactivity (*i.e.*, reactivity to either the MC2S domain or foldon domain), the serum was assayed against a Nipah F containing the respective tag.

## Cell culture, virus propagation and neutralisation assays

RSV A2 and MPV-GFP1 virus (strain CAN97–83) were propagated on Vero cells and stocks prepared were stored at -80°C. Viral titres were determined using plaque assay [58]. Briefly, Vero cells were seeded in 96-well tissue culture plates (Nunc) at $5 \times 10^4$ cells/well in Opti-MEM supplemented with 3% FBS and 100 U penicillin with 100 µg/mL streptomycin. After overnight growth to allow a confluent monolayer, serial dilutions of virus stock were prepared in Serum-free Opti-MEM with 100 U penicillin and 100 µg/mL streptomycin. It was also supplemented with 2 µg/mL of TPCK-Trypsin and 1X Glutamax in the case of MPV. Cell monolayers were inoculated with 50µL of virus and incubated for 1 hour at 37°C. After 1 hour incubation, in the case of RSV, the inoculum was topped up with overlay medium (Medium 199 with 1.5% methylcellulose, 2% FBS, 100 U penicillin with 100 µg/mL streptomycin), and in the case of MPV, the viral inoculum was removed before topping up with 1.5% CMC media. The plates were further incubated at 37°C for 72 hours. At the end of 72 hours, overlay medium was removed, the cells were fixed (80% acetone) and plaques were immunostained with either RSV specific human IgG Motavizumab or hMPV-specific human IgG MPV483. The primary antibody staining was followed by IRDye 800CW goat anti-human IgG, LI-COR Biosciences (1:2,500). Plaques were read using an Odyssey Imager (LI-COR Biosciences) and titre was recorded as PFU/mL.

PRNT assays were performed as previously described [58]. Briefly, mouse sera samples were heat-inactivated at 56°C for 30 minutes and serially diluted 1:10–1:21870. Equal volumes of viral suspension made in inoculum media by addition of virus stock to achieve the desired concentration of $2X10^3$ PFU/mL were added to the serum dilution to achieve a final

serum dilution of 1:20–1:43,740 and incubated at 37°C for 1 hour for the virus to be neutralised by serum antibodies. The inoculum media used for RSV was Opti-MEM with 100 U penicillin with 100 µg/mL streptomycin, while the media used for hMPV was also supplemented with 2 µg/mL of TPCK and 1X Glutamax. The residual viral infectivity in the sera/virus mixture was assayed by adding the mixture in duplicated to Vero cells monolayer cultivated in 96-well flat bottom plates. The plates were incubated at 37°C for 1 hour for virus adsorption to take place. This was directly followed by addition of overlay media in the case of RSV while the viral inoculum was removed before adding 1.5% CMC media in the case of hMPV. The plates were further incubated for 72 hours before fixing the cells with 80% acetone and plaque visualisation by immunostaining as above. PRNT assays for SARS-CoV-2 strain 614G and Influenza virus H1N1pdm A/California09 were conducted with similar methodologies as previously described [31,59]. Nipah virus neutralisation titre was assessed by lentivirus-based pseudo particle assay, as per a method previously described [60].

Human plasma depletion PRNT assays were performed as above. In the case of RSV, human plasma was serially diluted then first incubated at 37 °C for 1 hour either alone or with RSV Fsol, DSCav1 or VXB-213 diluted to 0.5 µg/mL. RSV A2 was then added to achieve the desired concentration of $2X10^3$ PFU/mL and again incubated at 37°C for 1 hour. The mixture of virus, human plasma and depletion antigen was then added directly to cells and the PRNT assay performed as above. In the case of hMPV, the presence of antigen within the human plasma depletion PRNT assay had a non-desirable effect of directly blocking viral entry, presumably by receptor binding. Instead, 1.4 mg of VXB-221 or DsCavEs2 were each coupled to 0.1 g of CNBR-Activated Sepharose resin (Cytiva), as per manufacturer's instructions. A control 0.1 g of CNBR-Activated Sepharose resin was produced with coupling buffer alone in parallel. The three Sepharose resins (control, VXB-221 and DSCavEs2) were inactivated and washed as per manufacturer's instructions and then equilibrated with PBS. 1 mL of heat inactivated (HI) human plasma was added to each Sepharose resin and incubated on an end over end mixer for 2 hours at room temperature. Antigen depleted and control depleted human plasma were then collected into a 1.5 mL microcentrifuge tubes and passed through a 0.22 µM filter syringe to ensure sterility and removal of Sepharose beads. Antigen depleted and mock depleted human plasma were then used directly in the hMPV PRNT assay as per mouse sera. Plaque visualisation for human plasma depletion PRNT was completed as above, however with RSV specific mouse IgG Motavizumab or hMPV-specific mouse IgG MPV483 followed by IRDye 800CW goat anti-mouse IgG, LI-COR Biosciences (1:2500).

## Statistical analysis

Analysis was performed using GraphPad Prism version 10.0.2 for Windows, GraphPad Software, San Diego, California USA, www.graphpad.com. Mann Whitney Test was performed to assess the statistical significance of differences between treatment groups in neutralising immune response. The limit of detection (LoD) for neutralisation assays is defined as half of the initial serum dilution factor. *Statistical significance is indicated as: $p < 0.0001$ (****), $p < 0.001$ (***), $p < 0.01$ (**), and $p < 0.05$ (*).*

## Supporting information

**S1 Fig. Reactivity of HIV+ reference serum with MC and MC2. (A)** NIBSC Reference serum (NIBSC code: 02/210). **(B)** BioRad Geenius HIV+ Serum (catalogue number #72460).
(TIF)

**S2 Fig. Size-exclusion chromatography (SEC) analysis of purified antigens.** (A) SARS-CoV-2 S MC and SARS-CoV-2 S MC2 assessed with a Superose 6 Increase 10/300GL column. (B) Nipah F MC and Nipah F MC2 assessed with a Superdex 200 Increase 10/300GL column. (C) Influenza A HA MC and Influenza A HA MC2 assessed with a Superdex 200 Increase 10/300GL column. (D) RSV F MC, RSV F MC2 and RSV F MC2S assessed with a Superdex 200 Increase 10/300GL column. Note: slight differences in elution volumes are likely due to inter-run variation and column compaction.
(TIF)

**S3 Fig. Virus neutralisation following vaccination.** Two intramuscular doses were administered to BALB/c mice three weeks apart with each 50µl dose containing either 5µg (SARS-CoV-2 S, Nipah F, Influenza HA), or 1µg (RSV F) of purified protein in PBS and 25µl of AddaVax (InvivoGen). Three weeks following the second dose, blood serum was collected for analysis of virus neutralisation. **(A)** SARS-CoV-2 prototypic Wuhan strain with D614G mutation assessed by $PRNT_{50}$ assay. **(B)** Nipah virus neutralizing titre assessed by lentivirus-based pseudoparticle assay. **(C)** Influenza H1N1pdm A/Cal09 neutralizing titre assessed by $PRNT_{50}$ assay. **(D)** RSV A2 neutralizing titre assessed by $PRNT_{50}$ assay Statistical analysis comparing neutralization between MC and MC2 by Mann-Whitney test (A-C), and statistical analysis comparing neutralization between MC, MC2 and MC2S, by Dunn's multiple comparisons test (D). Bars represent geometric means +/- Standard deviation.
(TIF)

**S4 Fig. Mouse sera IgG binding to RSV F ectodomain or clamp subdomain (MC, MC2 or MC2S) by ELISA. (A)** ELISA $EC_{50}$ values. **(B)** Relative percentage of IgG specific to clamp subdomain calculated via the formular $EC_{50}$ clamp subdomain (MC, MC2 or MC2S) divided by total elicited IgG response ($EC_{50}$ clamp subdomain plus $EC_{50}$ RSV F ectodomain). Bars represent geometric means +/- Standard deviation. ANOVA Dunn's multiple comparisons test with only significant values >0.05 shown.
(TIF)

**S5 Fig. Reactivity of mouse sera IgG binding to HIV-1 gp41 by ELISA. (A)** ELISA readings. **(B)** Relative endpoints. Bars represent geometric means +/- Standard deviation. A Mann Witney test used for statistical significance.
(TIF)

**S6 Fig. SDS-PAGE of RSV and hMPV antigens.** The antigens (5µg) were analysed by SDS-PAGE. All were run under reducing conditions, mixed with 100mM of dithiothreitol and 4X Laemmli sample buffer (BioRad) and boiled for 5 minutes. Gels were stained with Coomassie Brilliant Blue R-250 (BioRad) and destained using 50% RO water with 40% methanol and 10% acetic acid. To estimate the size of the denatured proteins, a Kaleidescope molecular weight ladder (BioRad) was included.
(TIF)

**S7 Fig. Ability of RSV and hMPV antigens to deplete neutralizing antibodies form human plasma. (A)** RSV $PRNT_{50}$ with titration of pooled human plasma (black) and human plasma pre-incubated with 0.5 µg/mL of RSV Fsol (orange), DSCav1 (blue), or VXB-213 (purple). **(B)** hMPV $PRNT_{50}$ with titration of pooled human plasma (black) or human plasma depleted of antigen-reactive antibodies via incubation with Sepharose immobilised antigens DsCavEs2 (blue) and VXB-221 (purple). Analysis was performed using GraphPad Prism version 10.0.2 for Windows, GraphPad Software, San Diego, California USA, www.graphpad.com.
(TIF)

**S8 Fig. Antigen Stability in liquid formulation. (A/B)** VXB-213 and VXB-221 Stability analysis out to 12 months at 2–8°C as determined by percentage of product as a single MW by reduced CE-SDS, SE-HPLC, or by potency testing capture ELISA with paired neutralising antibodies.
(TIF)

**S9 Fig. SDS-PAGE of RSV antigens.** The antigens (4 µg) were analysed by SDS-PAGE. All were run under reducing conditions, mixed with 100mM of dithiothreitol and 4X Laemmli sample buffer (BioRad) and boiled for 5 minutes. Gels were stained with Coomassie Brilliant Blue R-250 (BioRad) and destained using 50% RO water with 40% methanol and 10% acetic acid. To estimate the size of the denatured proteins, a Kaleidescope molecular weight ladder (BioRad) was included.
(TIF)

## Author contributions

**Conceptualization:** Paul R Young, Daniel Watterson, Keith J. Chappell.

**Investigation:** Andrew Young, Sharada Kolekar, Carlos Alverez Mendoza, Noushin Jaberolansar, Naphak Modhiran, Tim Webb, Robert McCuaig, Varsha Kommajosyula, Nicolas Tardiota, Quimbe Dy, Alberto A Amarilla.

**Project administration:** Rhiannon L Dalrymple, Marianne Gillard, Julie L. Dutton, Juana Magdalena, Frank Vandendriessche, Jean Smal, Paul R Young, Daniel Watterson, Emmanuel J. Hanon, Keith J. Chappell.

**Writing – original draft:** Andrew Young, Rhiannon L Dalrymple, Keith J. Chappell.

**Writing – review & editing:** Andrew Young, Sharada Kolekar, Carlos Alverez Mendoza, Noushin Jaberolansar, Naphak Modhiran, Tim Webb, Robert McCuaig, Varsha Kommajosyula, Nicolas Tardiota, Quimbe Dy, Alberto A Amarilla, Rhiannon L Dalrymple, Marianne Gillard, Julie L. Dutton, Juana Magdalena, Frank Vandendriessche, Jean Smal, Paul R Young, Daniel Watterson, Emmanuel J. Hanon, Keith J. Chappell.

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
