## [Decision Letter · Decision Letter 0]

A second-generation molecular clamp stabilised bivalent candidate vaccine for protection against diseases caused by respiratory syncytial virus and human metapneumovirus.

PLOS Pathogens

Dear Dr. Chappell,

Thank you for submitting your manuscript to PLOS Pathogens. After careful consideration, we feel that it has merit but does not fully meet PLOS Pathogens's publication criteria as it currently stands. Therefore, we invite you to submit a revised version of the manuscript that addresses the points raised during the review process.

Please submit your revised manuscript within 60 days Mar 29 2025 11:59PM. If you will need more time than this to complete your revisions, please reply to this message or contact the journal office at plospathogens@plos.org. Please include the following items when submitting your revised manuscript:

We look forward to receiving your revised manuscript.

Kind regards,

Alexander Bukreyev, Ph.D.

Academic Editor

PLOS Pathogens

Matthias Schnell

Section Editor

Editor-in-Chief

PLOS Pathogens

orcid.org/0000-0003-2946-9497

Michael Malim

Editor-in-Chief

PLOS Pathogens

orcid.org/0000-0002-7699-2064

**Journal Requirements:**

At this stage, the following Authors/Authors require contributions: Andrew Young, Sharada Kolekar, Carlos Alverez Mendoza, Noushin Jaberolansar, Naphak Modhiran, Tim Webb, Robert McCuaig, Varsha Kommajosyula, Nicolas Tardiota, Quimbe Dy, Alberto A Amarilla, Rhiannon L Dalrymple, Marianne Gillard, Julie Dutton, Juana Magdalena, Frank Vandendriessche, Paul R Young, Jean Smal, Daniel Watterson, Emmanuel Hanon, and Keith J. Chappell. Please ensure that the full contributions of each author are acknowledged in the "Add/Edit/Remove Authors" section of our submission form.

https://journals.plos.org/plospathogens/s/submission-guidelines#loc-parts-of-a-submission

3) We noticed that you used the phrase 'not shown' in the manuscript. We do not allow these references, as the PLOS data access policy requires that all data be either published with the manuscript or made available in a publicly accessible database. Please amend the supplementary material to include the referenced data or remove the references.

4) We do not publish any copyright or trademark symbols that usually accompany proprietary names, eg ©,  ®, or TM  (e.g. next to drug or reagent names). Therefore please remove all instances of trademark/copyright symbols throughout the text, including:

- ® on pages: 20 line 656, 21 lines 668 and 669, and 22 line 709

- TM on pages: 2 line 56, 4 lines 103, 104, 130, and 131, 7 line 216, 8 line 232, 12 lines 357, 360, 361, 368, 375, and 384, 13 lines 388, 390, and 413, 15 lines 455, 456, 460, and 462, 17 lines 528, and 530, and 19 lines 593, 594, 606, and 613.

5) Please upload all main figures as separate Figure files in .tif or .eps format. For more information about how to convert and format your figure files please see our guidelines: 

6) We have noticed that you have uploaded Supporting Information files, but you have not included a list of legends. Please add a full list of legends for your Supporting Information files after the references list.

7) We note that your Data Availability Statement is currently as follows: "All data are available within the manuscript and supporting information". Please confirm at this time whether or not your submission contains all raw data required to replicate the results of your study. Authors must share the “minimal data set” for their submission. PLOS defines the minimal data set to consist of the data required to replicate all study findings reported in the article, as well as related metadata and methods (https://journals.plos.org/plosone/s/data-availability#loc-minimal-data-set-definition).

8) Please amend your detailed Financial Disclosure statement. This is published with the article. It must therefore be completed in full sentences and contain the exact wording you wish to be published.

**Reviewers' Comments:**

Reviewer's Responses to Questions

**Part I - Summary**

Reviewer #1: In this work, Young et al. report a second-generation molecular clamp-stabilized bivalent RSV and hMPV vaccine candidate, VXB-241 (a combination of VXB-213 for RSV preF and VXB-221 for hMPV preF). This vaccine incorporates glycan-shielded molecular clamps (MC2S) and additional stabilizing substitutions for both RSV F and hMPV F proteins to maintain them in the prefusion conformation. The authors evaluated the immunogenicity of the MC2S-stabilized RSV and hMPV F antigens and compared neutralizing antibody titers as well as tag-specific antibody titers, demonstrating that their antigens elicited higher RSV/hMPV-specific antibody titers and lower tag-specific antibody titers than DsCav1 or DsCavEs2 (although they did not actually test DSCavEs2, see below). Furthermore, the authors conducted both research lab-scale and 50L industry-scale expressions and characterizations and monitored the stability of the designed antigens over nine months at 2–8 °C.

Overall, this manuscript reports research findings with therapeutic potential. However, there are concerns that need to be addressed, and the authors should improve the quality of their writing and figures. This includes addressing grammatical errors, incorporating statistical analyses into the plots, adding details to their methods, ensuring proper experimental controls, and citing appropriate literature to clarify their data interpretations.

Reviewer #2: In “A second-generation molecular clamp stabilised bivalent candidate vaccine for protection against diseases caused by respiratory syncytial virus and human metapneumovirus”, Young et al. develop a molecular clamp (MC2) that avoids use of HIV-1-gp41 sequences that might yield cross-reactive antigenicity with HIV-1 diagnostics. The authors further add N-linked glycans to create MC2S, and show reduced clamp-related immune responses. Then, using MC2S, RSV and hMPV candidate vaccines were generated along with benchmark comparisons from the field. For RSV, the prefusion stabilized VXB-213 was made that included both single chain modification and disulfide identified by Joyce et al. For hMPV, the prefusion stabilized VXB-221 was made, with proline and single disulfide.

Immunogens were assessed for antigenicity, purity, ability to deplete neutralizing antibodies from human sera, and assessed immunologically in BALB/c mice using a bivalent candidate vaccine, VXB-241. The authors observed similar RSV neutralizing titers as DSCav1, and slightly higher hMPV neutralizing titers as a disulfide stabilized comparator (DSCAvEs2).

The authors carry out detailed analysis of the yields for VXB-213 and VXB-221, which were robust at 1.7 and 5.5 g/L, respectively, with high antigen stability – and also repeat immunogenicity in mice.

Overall, I like the idea of making a tighter molecular clamp to stabilize the prefusion conformation. The authors may want to cite other stabilizations, such as the use of trimerization domains to stabilize EBV gB (https://www.biorxiv.org/content/10.1101/2024.10.23.619923v1).

Reviewer #3: The manuscript of Young et al., describes prefusion stabilized F proteins of RSV and HMPV, which are trimerized with a novel trimerization domain. The proteins are produced at high yield and show high stability. Mice immunization shows that MC2S stabilized RSV F (VXB-213) is superior to RSV DS-Cav1 and that VXB-221 is equivalent to HMPV DS-CavES2.

**Part II – Major Issues: Key Experiments Required for Acceptance**

Reviewer #1: 1. The authors state that their antigens are stabilized in the prefusion conformation due to the addition of MC2S (abstract: “Here we describe the design and characterisation as well as pre-clinical development of a bivalent vaccine, VXB-241, comprised of the recombinantly expressed viral fusion proteins from both RSV and hMPV, stabilised in their pre-fusion conformation by a second-generation molecular clamp (MC2S)”). However, the lead antigens all contain stabilizing amino acid substitutions described previously in the literature, including disulfide bonds. This is later clarified in the introduction, lines 146-150. Statements indicating that the clamp stabilizes the antigens in the prefusion conformation should be removed unless supporting data are provided.

2. Parts of the discussion read like an advertisement, such as the sentence “Here, we have demonstrated that the Molecular Clamp (MC) platform has significant potential for the development of a novel type of subunit vaccines offering several key advantages such as enhanced stability, improved immune responses, and the versatility to target diverse viral pathogens.” However, the authors have not directly demonstrated any advantage of the MC platform relative to other trimerization motifs. The lead antigens all contain stabilizing substitutions, which are required to stabilize the prefusion conformation. And at no point do the authors test a construct with identical stabilizing substitutions and different trimerizations motifs (MC2S vs Foldon, for instance). Therefore, statements regarding the advantages of the clamp should be considerably toned down, including “A transformative feature of MC2S…”.

3. The authors claim that they use DS-CavEs2 as a comparator for hMPV F, but they do not. DS-CavEs2 contains four disulfide bonds (L110C-N322C, T365C-V463C, T127C-N153C, A140C-A147C) along with substitutions A185P, V231I, L219K, and E453Q. However, the “DS-CavEs2” construct used by the authors (as defined in Table 1) contains only two disulfide bonds (365-463 and 127-153) along with A185P, L219K, and V231I. This construct is referred to as DS-CavEs in Hsieh et al 2022: “Furthermore, the variant containing all beneficial substitutions (DSx2/L219K/V231I), exhibited an additional 1.8-fold increase compared to DSx2 and had a Tm of 60.7 °C (Fig. 3d). We renamed this variant DS-CavEs…”. The authors should either test DS-CavEs2 or modify the text accordingly.

4. Figure 1E and Supplementary Figure 6: an explanation is needed for why DsCavEs2’s SEC trace and SDS-PAGE gel image are substantially different from the Hsieh et al, 2022 publication. Additionally, antigen QC data—such as thermostability and electron microscopy data—should be provided and compared with previous literature to confirm proper expression and folding of the antigens. A detailed protein purification method description, including affinity column types, brand names, catalog numbers, and buffer components, is also needed in the Method section.

5. Figure 3: Statistical analyses need to be added to the plots. Furthermore, for hMPV large-scale expression, the authors should incorporate previously established controls, such as DsCavEs2, as a reference.

6. Supplementary Figures 4 and 5 require statistical analyses to support the claim that MC2S induces immune silencing compared to MC2, with p-values and test details clearly presented. Additionally, the Methods section must describe the protein expression and purification procedures for the molecular clamps used in Fig S4 and HIV-1 gp41 used in Fig S5.

Reviewer #2: The major issue with this paper is the “best-in-class comparator antigens”. For RSV, the DSCav1 was developed more than a decade ago and is known to have less than optimal immunogenicity and stability (with significantly lower immunogenicity and stability than the 2nd-generation immunogens described by Joyce et al. 2016 NSMB); furthermore, DSCav1 fails when delivered by mRNA (Aliprantis et al. 2021 Hum Vaccin Immunother.) but succeeds when the additional stabilization described by Joyce et al. are incorporated. Thus, DSCav1 is not the best in class immunogens for RSV, and this needs to be corrected and discussed.

Similarly for hMPV, the comparator used is significantly less immunogenic than the triple disulfide versions, which were described in Ou et al. 2023 Plos Pathogens. Strangely, one of the most triple-disulfide comparators is shown in Table 1 (V3B-A12_D454C-V458C) but is this material is not compared immunogenically.

Bottom line: the author misrepresent their developed immunogens as being better than best-in-class comparators. The comparators chosen are not best-in-class. It would be optimal for the authors to carry out immunogenicity with best-in-class comparators.

Reviewer #3: The novelty of the manuscript is the use of a novel type of trimerization domain. This novel domain is likely another type of six-helix bundle of a non-human pathogen, but it is not described in the manuscript nor is there a reference to another paper. Therefore, this domain and the selection of the particular domain should be described in the paper. The design and the addition of glycans cannot be reviewed since there are no details.

The reduction of immune response against the silenced trimerization domain was two-fold. Although the response is lower compared with the non-silenced version, immune responses will increase after repeated vaccinations and the anti-MC2S will likely become more dominant after repeated vaccinations. Understandably, the authors want to highlight the novel trimerization domain, but they should not neglect earlier studies of pneumovirus vaccines without trimerization domains and include this in the discussion.

The authors write that comparable antigen homogeneity and oligomeric characteristics were conferred by MC and MC2 in Supplementary figure 2. However, in all cases except for influenza HA, the retention time is significantly shorter. How can this difference be explained? Has the Mw been analysed using SEC-MALS?

Why does the large SARS2 spike protein have a longer retention time compared with Nipah, RSV and influenza proteins?

Why is MC2 only superior to MC for the influenza HA?

The authors write that VXB-213 was engineered to include the optimal disulfide bridge linkages and furin cleavage site deletion identified by Joyce et al. Since this design is an optimized version of DS-Cav1, the authors should include text and reference to DS-Cav1 design (McLellan, 2013). VXB-213 also includes L373R. Could the author explain the origin and contribution of this mutation?

Line 208: VXB-221 does not include an (unmodified) furin cleavage site. It contains a TMPRSS2 site which is not cleaved in the expression system used.

The description of DsCavEs2 in Table 1 is incomplete. DsCavEs2 also includes disulfides 110-322, 140-147, and it contain mutation E453Q.

Could the authors explain this omission? Does it mean the description is wrong ? or is the naming wrong? In the last case, the authors have not compared immunogenicity of VXB-221 with a published variant, but with a protein of unknown characteristics and stability. The description of v3B_delta12 is correct and would have been a better comparator in the immunogenicity study based on the stability and other characteristics.

Table 1 describes that hMPV Fsol is a Postfusion control. However, there is no evidence that the soluble F is transformed to postfusion. Please include evidence or change description. The publication of Battles et. al., (2017) shows how a postF protein can be generated.

No superiority conclusions can be drawn from Ds-Cav1 and VXB-213 antigens, nor the DsCavEs2 vs VXB-221 antigens based on the ELISA or the mouse immunogenicity studies. Especially because the purification strategy does not allow such conclusions. The relatively unstable DsCav1 is purified using immunoaffinity strategies for which relatively harsh elution conditions are needed. Moreover, elution from 101F is not similar as elution from AVI8740. Apart from the sensitivity of the prefusion stability, based on supplementary Figure 6 (move this figure to main figures), the authors have not used the optimal purification method for DsCav1 and DsCavEs2 based on impurities shown in the gel, especially for DsCav1. Animal experiments can only be compared if purity is similar. Proteins should have been purified with additional SEC purification.

Moreover, if authors want to compare the vaccines, a dose titration study should be done instead of just one concentration.

It is the stabilizing mutations that determine the stability of the prefusion conformation which is important for vaccine efficacy. Since for RSV and HMPV preF, (Ds-Cav1 vs VXB-213 and DsCavEs2 vs VXB-221) the stabilization strategy and the purity is different, it is not possible to draw conclusions on trimerization domain.

Since the novelty of the manuscript is the novel trimerization domain, not only the purity of the vaccines should be the same, but also the stabilizing substitutions should be identical between the foldon-fused and the MC2S-fused immunogens.

Line 277: why was hMPV Fsol not used for depletion studies? And how is it possible that one animal immunized with Fsol has the highest PRNT50 titer of all animals?

Line 351: DS7 is a bad choice if the authors want to catch the native prefusion conformation since DS7 recognizes a postF epitope.

For the comparison of VXB-213 with Arexvy, the lyophilized proteins was reconstituted with PBS containing 0.05% PS80. For Arexvy, the adjuvant is part of the diluent, and from the description of Abrysvo it appears to be water. So both vaccines are not dissolved according to the instructions for an optimal formulation.

Were the immunisations with Arexvy performed with correctly reconstituted vaccine including the ASO1B adjuvant? Or was it PBS/tween reconstituted and mixed with Addavax?

Line 441: Furthermore, the second generation MC2S also enhances a variety of antigen characteristics, such as conformational stability and yield relative to the first-generation MC (data not shown).

If not shown and no reference, this line should be deleted.

The comment in line 481 that VXB-213 could elicit higher neutralization at lower dosage than a DsCav1-based vaccine is speculation. Pfizer's Pref 847 also showed a much stronger induction of VNT compared with DS-Cav1. However, human trials showed same level of protection.

Combination of vaccines will indeed be an important next step in respiratory vaccine development. It is not clear why the stable components of Abrysvo would not be suited for such combinations.

Since vaccines will be used in pre-immune population, a comparison in a pre-immune model would be interesting.

The authors claim that the neutralizing antibodies against the internal trimer interface are potent. However, after the publication of Rapazzo et. al. (Immunity 2022), such antibodies can no longer be labeled as potent. Site 0 antibodies like ADI-61026 are now the benchmark of potent neutralizing antibodies.

References to the Icosavax press release don’t work. I am also not sure whether references to press releases should be used for scientific manuscripts.

For antigen – specific IgG responses, the antigens are coated directly on ELISA plates. This can cause denaturation and loss of epitopes.

In line 510, a reference is made to a fusion inhibitor study of Battles et. al., (2016). This reference seems out of place. The Krarup (Nat Comm 2015) publication describes the dependency on full cleavage for RSV F trimerization.

Line 511 and beyond has too much speculation on the possible advantages of an open HMPV trimer vaccine and lacks discussion of recent insights.

**Part III – Minor Issues: Editorial and Data Presentation Modifications**

Reviewer #1: 1. Line 62: RSV was discovered in 1956, not 1957.

2. Supplementary Figure 2: The authors should provide an explanation for the shifted peaks observed between MC and MC2 antigens in panels A, B, and D, but not C. Clarify whether these shifts are expected or indicative of structural or conformational differences.

3. Supplementary figure 3: Specify which dose(s) was/were used for animal immunizations. Currently, the description in line 626 ("Mice were immunized with 0.5 µg, 2 µg, or 3 µg per recombinant protein antigen stated") is unclear and makes interpretation of results difficult.

4. Lines 219-220: the sentence “RSV preF antigens, 94% aggregate, 0% trimer; hMPV preF antigen, 25% aggregate, 19% trimer, 56% monomer” is very vague and confusing. Please clearly specify which antigens are being discussed within the paragraph.

5. The term "dissociation constant" is incorrectly typed as “kD” throughout the manuscript. Please replace with “KD”.

6. Line 285: the authors should clarify that “VXB-241” is a combination of VXB-213 and VXB-221 antigens.

7. Line 377: Based on Figure 3B, value “15.5%” seems to be a typo, as the values for Arexvy antigens in the plot are 19.1% and 19.2%. The authors should check for data accuracy.

8. Line 398–399: The statement “…potentially indicating some interference due to immunodominance of the RSV preF antigen” requires further clarification. Either provide supporting data or cite relevant literature, as there is currently no evidence included in the figures to substantiate this claim.

Reviewer #2: (No Response)

Reviewer #3: (No Response)

PLOS authors have the option to publish the peer review history of their article (what does this mean? ). If published, this will include your full peer review and any attached files.

**Do you want your identity to be public for this peer review?** For information about this choice, including consent withdrawal, please see our Privacy Policy .

Reviewer #1: No

Reviewer #2: No

Reviewer #3: No

**Figure resubmission:**

**Reproducibility:**



---

## [Decision Letter · Decision Letter 1]

A second-generation molecular clamp stabilised bivalent candidate vaccine for protection against diseases caused by respiratory syncytial virus and human metapneumovirus.

PLOS Pathogens

Dear Dr. Chappell,

Thank you for submitting your manuscript to PLOS Pathogens. After careful consideration, we feel that it has merit but does not fully meet PLOS Pathogens's publication criteria as it currently stands. Therefore, we invite you to submit a revised version of the manuscript that addresses the points raised during the review process.

Please submit your revised manuscript within 60 days Jul 19 2025 11:59PM. If you will need more time than this to complete your revisions, please reply to this message or contact the journal office at plospathogens@plos.org. Please include the following items when submitting your revised manuscript:

We look forward to receiving your revised manuscript.

Kind regards,

Matthias Johannes Schnell, PhD

Section Editor

PLOS Pathogens

Editor-in-Chief

PLOS Pathogens

Editor-in-Chief

PLOS Pathogens

orcid.org/0000-0002-7699-2064

**Journal Requirements:**

At this stage, the following Authors/Authors require contributions: Andrew Young, Sharada Kolekar, Carlos Alverez Mendoza, Noushin Jaberolansar, Naphak Modhiran, Tim Webb, Robert McCuaig, Varsha Kommajosyula, Nicolas Tardiota, Quimbe Dy, Alberto A Amarilla, Rhiannon L Dalrymple, Marianne Gillard, Julie Dutton, Juana Magdalena, Frank Vandendriessche, Jean Smal, Paul R Young, Daniel Watterson, Emmanuel Jules Hanon, and Keith J. Chappell. Please ensure that the full contributions of each author are acknowledged in the "Add/Edit/Remove Authors" section of our submission form.

- ® on page: 25 line 744.

3) Please amend your detailed Financial Disclosure statement. This is published with the article. It must therefore be completed in full sentences and contain the exact wording you wish to be published.

1) If the funders had no role in your study, please state: "The funders had no role in study design, data collection and analysis, decision to publish, or preparation of the manuscript."

**Reviewers' Comments:**

Reviewer's Responses to Questions

**Part I - Summary**

Reviewer #1: The authors have generally addressed my prior comments, and changes to the resulting text have improved the accuracy of those statements. It is a pity though that comparisons to previously reported antigens, since as Ds-CavEs2, are still difficult to interpret given that the authors did not express and purify the antigens according the published procedures. There is presumably a reason why co-transfection with a plasmid containing furin and SEC purification was performed in Hsieh et al. At least now the authors of this manuscript acknowledge that some of the differences observed in their studies could be due to different expression and purification strategies.

Reviewer #2: Chappell and colleague describe an interesting trimeric molecular clamp, derived the post-fusion HIV-1 gp41 six-helix bundle, which they utilize in vaccines against respiratory syncytial virus and human metapneumovirus. The manuscript has a lot of nice data, but has suffered from “overclaiming”, as indicated by comments from the reviewers. These substantially detracted from an otherwise nice paper.

Reviewer #3: -

**Part II – Major Issues: Key Experiments Required for Acceptance**

Reviewer #1: N/A

Reviewer #2: (No Response)

Reviewer #3: The authors have included acknowledgement that differences in the methods used to comparator antigens is a potential limitation, but they only describe this for the HMPV immunogen and not for the 101F based purification of RSV immunogens. The limitation of the purification of the latter should also be included irrespective of the fact that another experiment was done with the commercial vaccine as a comparator.

The comment that VXB-213 could elicit higher neutralization at lower dosage than a DsCav1-based vaccine is speculation. The authors agree and mention that the clinical trial is underway but they did not delete the speculative sentence. This should be deleted and authors should wait for the trial data to make this claim.

Reviewer mentioned that in line 511 and beyond (first version) there was too much speculation on the possible advantages of an open HMPV trimer vaccine and lacks discussion of recent insights.

Authors agree that it is speculative but decided not to change the text while this section is still too speculative and should be toned down. If the authors insist to discuss so much details on this subject and if they think it is relevant, they should mention that the anti-interface antibodies are far less potent compared with the site 0 antibody ADI-61026

**Part III – Minor Issues: Editorial and Data Presentation Modifications**

Reviewer #1: N/A

Reviewer #2: While the authors have done a decent job responding to reviewers, and the manuscript is improved, there are still overstatements that should be removed.

For example, in the abstract, the authors have not modified their best-in-class comparative claim that “Head-to-head evaluations in mouse immunogenicity studies showed that the VXB-241 candidate vaccine induced a neutralizing immune response that was superior or equivalent to the best-in-class comparator antigens…” As indicated in my prior review, it’s inappropriate to call the DS-Cav1 immunogen a “best-in-class comparator antigen” when discussing immunogenicity, as there are other described antigens with higher immunogenicity, including those from Joyce et al, with more than 3-fold increased immunogenicity in mice than DS-Cav1.

The authors counter in their response that DS-Cav1 is the best-in-class, since the clinical trial results are so good. But what is being compared in this paper is mouse immunogenicity, not clinical trial results. Thus, it’s critical that the authors do not call DS-Cav1 ‘best-in-class’, when referring to immunogenicity data. In truth, there wouldn’t be much difference to the abstract to merely removed the “best-in-class” state, e.g., “Head-to-head evaluations in mouse immunogenicity studies showed that the VXB-241 candidate vaccine induced a neutralizing immune response that was superior or equivalent to comparator antigens…” This accurately states the data but does not overclaim.

Also, since the MC2S is the focus of this paper, it might be helpful to have a figure showing the explicit modifications that were introduced; such a figure could easily be added to the supplemental, enabling interested readers to immediately see this important part of the manuscript.

Lastly, since MC2S is a trimerization “clamp” based on helical interactions, the authors should add a few sentences to the discussion comparing MSC2 to other trimeric stabilizations, such as the trimer-adapted version of GNC4, which as been used to stabilized other prefusion trimeric immunogens as well as the helical zipper derived from RSV itself (Stewart-Jones, et al., 2015 “A Cysteine Zipper Stabilizes a Pre-Fusion F Glycoprotein Vaccine for Respiratory Syncytial Virus”).

Reviewer #3: -

PLOS authors have the option to publish the peer review history of their article (what does this mean? ). If published, this will include your full peer review and any attached files.

**Do you want your identity to be public for this peer review?** For information about this choice, including consent withdrawal, please see our Privacy Policy .

Reviewer #1: No

Reviewer #2: No

Reviewer #3: No

**Figure resubmission:**
---

## [Editor Report · Decision Letter 2]

A second-generation molecular clamp stabilised bivalent candidate vaccine for protection against diseases caused by respiratory syncytial virus and human metapneumovirus.

PLOS Pathogens

Dear Dr. Chappell,

Thank you for submitting your manuscript to PLOS Pathogens. After careful consideration, we feel that it has merit but does not fully meet PLOS Pathogens's publication criteria as it currently stands. Therefore, we invite you to submit a revised version of the manuscript that addresses the points raised during the review process.

Please submit your revised manuscript within 60 days Jul 28 2025 11:59PM. If you will need more time than this to complete your revisions, please reply to this message or contact the journal office at plospathogens@plos.org. Please include the following items when submitting your revised manuscript:

We look forward to receiving your revised manuscript.

Kind regards,

Matthias Johannes Schnell, PhD

Section Editor

PLOS Pathogens

Matthias Schnell

Section Editor

PLOS Pathogens

Editor-in-Chief

PLOS Pathogens

orcid.org/0000-0003-2946-9497

Editor-in-Chief

PLOS Pathogens

**Journal Requirements:**

At this stage, the following Authors/Authors require contributions: Noushin Jaberolansar. Please ensure that the full contributions of each author are acknowledged in the "Add/Edit/Remove Authors" section of our submission form.

2) Please ensure that the funders and grant numbers match between the Financial Disclosure field and the Funding Information tab in your submission form. Note that the funders must be provided in the same order in both places as well. State the initials, alongside each funding source, of each author to receive each grant. For example: "This work was supported by the National Institutes of Health (####### to AM; ###### to CJ) and the National Science Foundation (###### to AM).".

**Reviewers' Comments:**

**Figure resubmission:**

**Reproducibility:**



---

## [Editor Report · Decision Letter 3]

Dear Dr Chappell,

We are pleased to inform you that your manuscript 'A second-generation molecular clamp stabilised bivalent candidate vaccine for protection against diseases caused by respiratory syncytial virus and human metapneumovirus.' has been provisionally accepted for publication in PLOS Pathogens.

Best regards,

Matthias Johannes Schnell, PhD

Section Editor

PLOS Pathogens

Matthias Schnell

Section Editor

PLOS Pathogens

Sumita Bhaduri-McIntosh

Editor-in-Chief

PLOS Pathogens

orcid.org/0000-0003-2946-9497

Michael Malim

Editor-in-Chief

PLOS Pathogens

orcid.org/0000-0002-7699-2064
---

## [Editor Report · Acceptance letter]

Dear Dr Chappell,

We are delighted to inform you that your manuscript, "A second-generation molecular clamp stabilised bivalent candidate vaccine for protection against diseases caused by respiratory syncytial virus and human metapneumovirus.," has been formally accepted for publication in PLOS Pathogens.

Best regards,

Sumita Bhaduri-McIntosh

Editor-in-Chief

PLOS Pathogens

orcid.org/0000-0003-2946-9497

Michael Malim

Editor-in-Chief

PLOS Pathogens

orcid.org/0000-0002-7699-2064